# Deaminase Modulation Driving a New Era in Drug Development

**DOI:** 10.3390/ijms262311532

**Published:** 2025-11-28

**Authors:** Robyn A. Lindley

**Affiliations:** 1Department Clinical Pathology, Victorian Comprehensive Cancer Centre (VCCC), Faculty of Medicine, Dentistry and Health Science, University of Melbourne, Melbourne 3052, Australia; robyn.lindley@unimelb.edu.au; 2GMDx Group Ltd., Melbourne 3123, Australia

**Keywords:** drug development, immunity, inflammation, cancer, chronic diseases, prognostication, endogenous mutagenic deaminases, AID cytosine deaminases, APOBEC cytosine deaminases, ADAR adenosine deaminases, small molecule deaminase inhibitors, functional modulation and immunotherapy

## Abstract

Our expanding understanding of the complex roles of endogenous mutagenic deaminases in human disease is driving the development of a new generation of therapeutics. These emerging drugs aim to achieve clinical benefit by modulating deaminase activity. Because these enzymes are intrinsic to key inflammation-related pathways, they represent promising targets for future therapeutic innovation. Although only a small number of deaminase-modulating agents have been approved for clinical use, many more are currently under investigation. Here, we present examples that illustrate the therapeutic potential of modulating this diverse family of enzymes and identify some of the challenges and opportunities that warrant further exploration.

## 1. Aim of This Review

We are now becoming aware of the crucial immunomodulatory and oncogenic roles that a group of endogenous mutagenic enzymes called deaminases play in health and disease. All metazoan life forms carry a cargo of cytosine (C) and adenosine (A) deaminases that co-ordinate a wide range of genomic, epigenomic and regulatory modifications that are a crucial part of an inflammatory response [1,2]. [Box 1 provides a summary of core deaminase concepts]. The aim of this review is to describe how these enzymes are now being repurposed as precision therapeutic tools fueling the development of a fundamentally new generation of deaminase modulation drugs. While only a few deaminase modulating drugs are currently approved for clinical use, many are in development. We use examples to highlight the therapeutic promise of these new drugs.

Box 1Summary of core deaminase concepts (Source: Lindley 2020 [1]).The endogenous mutagenic deaminases are enzymes that catalyze the removal of an amino (-NH2) group from nucleobases in DNA or RNA, leading to base conversions that can introduce mutations. Deaminases play pivotal roles in many inflammation-linked diseases and cancers [1]. In humans, the 15 different deaminase proteins form two families:i. Cytidine deaminases: Cytidine deaminase editing is a hydrolytic deamination reaction causing C-to-U (U-uracil) changes and resulting in mutations of Cs predominantly on single stranded DNA (ssDNA). The cytidine deaminase family consists of Activation-Induced Deaminase (AID) and APOlipoprotein B mRNA Editing Catalytic polypeptides (APOBECs1,2,3A,3B,3C,3D, 3E, 3F,3G, 3H and APOBEC4).ii. Adenosine deaminases: Adenosine deaminase editing is a hydrolytic deamination reaction causing A-to-I (I-inosine) changes and resulting in mutations of As predominantly on double stranded RNA (dsRNA). This transcription-linked process diversifies a plethora of transcripts, including coding and noncoding RNAs. Current analyses suggests that these transcriptome mutations are also often copied back to the DNA of the cancer genome [3]. The adenosine deaminase family consists of Adenosine Deaminases Acting on RNAs 1, 2 and 3 (ADARs 1 and 2; and ADAR3 that is designated as enzymatically inactive).Some deaminases target only a single tissue type or tissue group while others such as AID and APOBEC1 are ubiquitous. Each member of the deaminase family has evolved to be different across species. There is some indirect evidence that many of the mendelian genetic variations across human sub-populations may be due to previous deaminase activity during responses to pathogenic infections in humans over evolutionary time [4].Deaminase mutation sites are defined by the type of mutation (e.g., C-to-T) and they preferentially target key motifs (e.g., for AID it is WRC where W = A/T, R = A/G and C is the mutated nucleotide). For catalytic reasons, deaminase mutations also occur in codon context (i.e., preferentially deaminating either the first, second or third nucleotide in the 3N codon structure read in the 5-prime (5′) to 3-prime (3′) direction [5].Different deaminases encode inferred deaminase binding domains (infDBDs) to catalytically bind to specific nucleotide sequences (‘motifs’) on ssDNA, ssRNA, dsRNA or RNA:DNA substrates that normally only occur during transcription and reverse transcription (RT).

## 2. Deaminase Action in Inflammation, Immunity and Disease

Inflammation is part of the body’s immune defense that is activated in response to foreign pathogens (e.g., viruses, bacteria, prions, fungi), environmental agents (e.g., toxins, UV exposure, physical trauma) and imperfect wound healing [6]. Figure 1 provides a schematic representation of the causal links between an inflammatory response and the roles of deaminases in immunity and inflammation-linked disease progression. During a normal inflammatory response, a complex set of interferon-stimulated gene (ISG) pathways are activated. Some deaminase genes are encoded by the ISGs, with their transcription upregulated when interferon signaling is triggered [7]. Deaminase catalysis is essential for both innate and adaptive immunity. In the innate immune system, many deaminases directly mutate the DNA or RNA of foreign pathogens to attenuate or eliminate the level of viable pathogenic progeny. In the adaptive immune system, B lymphocyte activation induced cytidine deaminase (AID) activity plays crucial roles in somatic hypermutation (SHM) and class switch recombination (CSR) for generating both the diversification of the functional class of immunoglobulins (Ig) and by enhancing antibody specificity and affinity.

The activity or dysregulation of deaminase proteins may also result in some uncorrected mutations in normal somatic tissue during cellular transcription. While most mutations are corrected, the accumulation of uncorrected mutations is now causally linked to the progression of many inflammation-linked diseases, aging and oncogenesis. Figure 2 shows the key molecular steps involved in deaminase-driven transcription and reverse transcription (DRT) activity [3,5] and shows when single stranded DNA (ssDNA), double stranded RNA (dsRNA) and annealed nascent RNA:DNA hybrids become available for deaminase mutagenic activity.

## 3. Innate Immune Response

The AID, APOBEC3s (apolipoprotein B mRNA-editing enzyme, catalytic polypeptide 3s) and ADARs (adenosine deaminases acting on RNA) are crucial parts of the first line of innate immune defense against invading pathogens. They act coordinately to reduce the number of viable progeny or to eradicate the pathogenic impact on the host by launching a direct mutagenic attack on various DNA or RNA target motifs exposed during the lifecycle of most pathogens. As many of the pathogenic progeny may remain viable with the accumulated new mutation burden, the pathogen may in turn benefit from some mutations and acquire altered characteristics to be passed on as a new strain to another host of the same species, or from vertebrate non-humans to humans as a zoonotic disease. This host–pathogen battle may accelerate the evolution of new pathogenic or more virulent strains and possibly promote drug resistance such as antibiotic resistance.

### 3.1. APOBEC3 Deaminases as ‘Viral Smashers’

The human APOBEC3 enzymes have been widely studied and have been shown to impact replication of many viruses such as human immunodeficiency (HIV), hepatitis B virus (HBV), flaviviruses such as Zika [10], coronaviruses like severe acute respiratory syndrome coronavirus 2 (SARS-CoV-2) [11,12], herpesviruses, and papillomaviruses, and several retroelements [13]. The role of deaminases in building innate immunity for HBV has been widely studied. It is a major risk factor predicting the likelihood of developing hepatocellular carcinoma (HCC) [14,15]. The cytidine deaminases suppress HBV replication by deaminating and destroying the major form of the HBV genome, the covalently closed circular DNA (cccDNA), without toxicity to the infected cells [16,17]. Interferon (IFN)-alpha is used to indirectly activate the APOBEC3 genes in primary hepatocytes [18] to suppress HBV transcription and replication, and there is also a lower risk of cancer occurrence [19].

Thus, in recent years, engineering APOBEC3 deaminases has emerged as a promising strategy in antiviral research. Several examples show that even minor changes in the structure of deaminases can give them completely new and unique properties [20]. Although, at present, the development of such antivirals is complicated by the lack of tools for activating and controlling deaminase expression in vivo, this strategy presents an opportunity to design new ways to enhance our innate immunity in the face of evolving or emerging new viruses.

### 3.2. Using Deaminases to Develop New Antimicrobial Drugs

The emergence of antimicrobial drug resistance is a significant global threat leading to a rapid increase in the prevalence of bacterial infections that can no longer be treated with available antibiotics. The World Health Organization estimates that by 2050 up to 10 million deaths per year might be attributed to antimicrobial resistance [21]. The same host–pathogen interactions that give rise to new viral strains can also be responsible for the evolution of microbes that become resistant to current antibiotics.

At present, two structure-based drug design approaches are being used to fight antibiotic resistant microbes: these include targeting structures within bacterial cells (similar to existing antibiotics) and/or targeting virulence factors rather than bacterial growth [21]. Some of these new drug development strategies are now being used in preclinical studies to identify new molecular candidates for further investigation in animal and human trials. Several advancements in deaminase engineering also hold potential for introducing precise genomic modifications in microbes to counteract antibiotic resistance mechanisms. In recent years, engineered tRNA-specific adenosine (TadA) deaminase variants have been widely studied and adapted for base editing in genome engineering. TadA deaminase variants, enhanced cytosine base editors have been created with high on-target activity and reduced off-target effects [22,23].

An alternate deaminase-modulating drug development strategy could involve engineering targeted nuclear-localized DBD heterodimers to build a new generation of microbe-specific antimicrobials. As an example, deaminases such as engineered ADAR1p110 could be used as a nuclear shuttle vector to deliver the DNA and/or RNA ‘nuclear microbe warheads’ carrying a payload of microbe-specific DBDs to their intended targets and catalytically rendering them unviable [20]. Single amino acid substitutions in deaminases can lead to changes in their recognized nucleic acid type, or cause deaminases to accept either DNA or RNA targets.

## 4. Adaptive Immune Response

In the adaptive immune system, AID is essential for generating new antibody repertoires to effectively fight foreign pathogens [24]. AID was the first cytosine deaminase to be identified by Tasuku Honjo’s group in 1999 when it was shown to be essential for SHM and CSR in both mice and humans [25,26]. The DNA deamination model of how AID promotes antibody diversity among immunoglobulin (Ig) genes is now widely supported [27,28]. As the efficacy of our adaptive immune system is dependent upon AID proteins, the regulation of AID expression in B-cells is important. The therapeutic enhancement or suppression of AID might therefore be clinically beneficial to modulate AID enzymatic activity in lymphoid tissue. It is a curious anomaly that there is still only indirect evidence suggesting that the roles of both APOBECs and ADARs are important in promoting B cell diversification [5,29,30].

### 4.1. Therapeutic Inhibition of AID in Lymphoid Tissue

A wide range of adverse health conditions such as chronic inflammation are associated with AID dysregulation or overexpression. Abnormalities in AID have been shown to disrupt gene networks and signaling pathways in both B-cell and T-cell lineage lymphoblastic leukemia, although the full extent of its role in lymphoid carcinogenesis remains unclear [31]. A classic manifestation of downregulation of AID expression is hyper-IgM syndrome (HIGM2) [26]. The most direct manifestation of the upregulation of AID expression in lymphoid cells is that it promotes the development of B-cell lymphomas that emerge from hypermutation and antigenic selection in post-antigenic germinal centers (GCs). This is due to its natural ability to modify DNA through deamination in lymphoid cells [32,33]. Lymphomas can result from AID causing strand breaks in the Ig heavy chains regions [34,35]. Montamat-Sicotte et al. [36] found that Heat Shock Protein 90 inhibitors decrease AID protein levels and reduced disease severity in a mouse model of acute B-cell lymphoblastic leukemia in which AID accelerates disease progression. Tepper et al. [37] showed that poly(ADP-ribose) polymerase-1 (PARP-1) is another key factor restricting AID activity and SHM at the Ig variable region. Thus PARP-1 inhibitor therapy might also be used more widely as an AID inhibitor. Other consequences of AID overexpression include AID-generated mutations that result in resistance to lymphoma drugs [38]. Thus, there is mounting evidence pointing to the possible clinical benefits of therapeutically suppressing AID for a range of B-cell related diseases.

In 2015, King et al. [39] demonstrated that AID activity can be modulated by modifying the accessibility of the deamination catalytic pocket and the DNA-binding ability in its tertiary structure. More recently, to develop the first generation of AID small molecules that could modulate AID activity, Alvarez-Gonzalez et al. [40] used a fluorescence-based reporter assay to identify three compounds that reduced CSR and base editing and thus demonstrating the feasibility of modulating AID activity in biological systems. Such studies are laying an important foundation for the eventual development of this new drug class of AID inhibitors.

### 4.2. Can AID Be Used to Improve Vaccine Efficacy?

There is a need to improve the efficacy of vaccines by investigating new adjuvant therapies to boost immunogenicity. At present, aluminum salt adjuvants like aluminum hydroxide gel and aluminum phosphate are often used as commercial antigen-nonspecific vaccine adjuvants to potentiate the immune response [41]. These work by inducing a 2-3-fold higher rate of B cell mutations that are associated with a higher expression of AID, which is the major enzyme controlling B cell receptor (BCR) affinity maturation [42].

Many studies have also been conducted to investigate the use of interferons as adjuvants (IFN-α, -β, -γ and -λ) [43]. Recently there has also been an increased interest in using self-assembled nanoparticles based on graphene oxide (GO) quantum dots as a vaccine adjuvant. As an example, zinc-based graphene oxide adjuvant reagent (ZnGC-R) is designed as a new adjuvant for the influenza vaccine. It has a higher been shown that ZnGC-R induced a 342% stronger IgG antibody response compared with vaccines using inactivated virus alone and leading to 100% in vivo protection efficacy against an H1N1 influenza virus challenge [44]. However, GO injections can lead to significant oxidative stress and inflammation, as their use stimulates the release of a specialized subset of cytokines that primarily direct cell migration [45]. The antigen-presenting cells migrate to the lymph nodes, where antibody class switching and maturation take place, giving rise to high-affinity and long-lasting antibody production (Figure 1) [46].

Each of these current approaches for developing new adjuvant therapies relies on indirect strategies to potentiate immunogenicity. As deaminase proteins are expressed as a subset of the IFN-stimulated cytokine release pathway, the translation and optimization of AID-based adjuvants might therefore be considered for future investigation.

## 5. Cytidine Deaminase Modulation and Non-Lymphoid Cancer

Chronic inflammation is a known risk factor for cancer initiation and progression [47]. Once oncogenesis is initiated, deaminase mutagenic activity becomes increasingly “dysregulated” by potentially targeting expressed genes during transcription. In this section, some examples reporting the diverse roles of deaminases in oncogenesis and the emerging opportunities for developing new deaminase-based therapies are identified.

### 5.1. AID Modulation in Non-Lymphoid Cancers

Apart from AID’s natural physiological function in B cell maturation and antibody production, AID also introduces double-strand breaks in non-Ig genes making them more unstable [3,4] and associated with chromosomal translocations [33,48]. Some AID-driven mechanisms in non-B cells have been identified as oncogenic [31,32,49] and also play a role in the reprogramming of genomic DNA methylation [50,51,52]. Gene promoters targeted by AID exhibit abnormally low methylation levels, indicating high activation of these genes [53]. It has also been reported that the tumor necrosis factor a (TNFα) triggers abnormal AID expression in certain inflammation-related cancers such as helicobacter pylori-associated gastric cancer and colitis-associated colon cancers [54,55].

Our understanding of these and other roles of AID in non-B cell cancers is further compounded by the increasing evidence showing that there is a high level of Ig expression in many non-lymphoid malignancies and that it is produced by the cancer cells themselves. Although the cancer-derived Ig shares identical basic structures with B cell-derived Ig, the cancer-derived Ig has restricted variable region sequences and shows aberrant glycosylation [56]. Additionally, cancer-derived IgG was shown to be a predictor of lymph node metastasis and worse prognostic outcomes. However, although AID has a direct mutagenic role in cancers, we have little understanding of the co-dependence of the roles of AID in SHM and non-B cell Ig expression in tumors. This suggests that it is likely to be some time before new AID modulating drugs will be approved for clinical use in non-lymphoid cancers.

### 5.2. APOBEC3 Modulation in Oncology

Over the last decade, many studies have contributed to our understanding of how members of the APOBEC3 homologous family of deaminases influence oncogenesis. The APOBEC3-mediated signatures are often detected in sub-clonal branches of tumor phylogenies and are acquired in cancer cell lines over time, indicating that APOBEC3 mutagenesis is likely ongoing in oncogenesis [57]. Since APOBEC3s induce a high level of DNA damage, their targeted overexpression in cancer cells may also be cytotoxic [58]. Based on these and many other APOBEC3 studies, it is evident that there are many new development opportunities using APOBEC3 modulation in cancers (Table 6 [59]). The following examples draw attention to how some of these may improve clinical outcomes.

#### 5.2.1. Promoting Immune Activation in the Tumor Microenvironment

Pan-cancer analyses have found that high APOBEC3-mediated mutagenesis is associated with increased immune activation in the solid tumor microenvironment for a range of cancers [60]. For example, studies in breast cancer report that high APOBEC3B expression is associated with more tumor infiltrating lymphocytes [61,62]. APOBEC3C-H expression levels are correlated with more cytotoxic T cell lymphocyte (CTL) effector-form CD8 T cells in the tumor microenvironment, increased T cell receptor diversity, and greater cytolytic activity [62,63]. Similar immune activation has been observed in bladder cancer, with multiple studies detecting increased immune signatures and interferon signaling in APOBEC3-high tumors [64,65]. In lung cancer, T cell-mediated immune activation was linked with high APOBEC3B expression and high APOBEC3 induced mutation loads [66,67]. In ovarian cancer, elevated APOBEC3B and APOBEC3G expression have been associated with greater immune cell infiltration [68].

A few studies have also reported that APOBEC3s are immune-suppressive (Table 4 [59]). In some cancers a higher APOBEC3B expression was found to be associated with less immune cell infiltration in adrenocortical carcinoma and gastric cancer [60,69]. Although not in the tumor itself, increased APOBEC3A expression causes C-to-U mutations in RNA in thousands of genes and in monocytes and macrophages [70], and it has been shown to shift macrophage polarization to a pro-inflammatory, immune-activating state [71,72]. Activation of APOBEC3A alone may also cause apoptosis in vitro, as it has the highest deamination activity among the APOBEC3s [73,74].

Understanding the relationship between APOBEC3B activity and cyclic hypoxia in the tumor microenvironment is also important. Evidence from in vivo experiments suggests that cyclic hypoxia is the primary cause of most deaminase-driven solid tumor cancers and is associated with the resistance to standard therapies, genomic instability and a poor patient prognosis [75]. Using tumor cell lines for a number of cancers that included colorectal, breast, bladder, lung, and esophageal tumor cell lines, it was shown that cyclic hypoxic conditions induce the expression and activity of APOBEC3B.

In summary, controlled APOBEC modulation, rather than complete inhibition in the tumor microenvironment of some cancers, could improve patient outcomes by promoting tumor immunogenicity. Other impacting factors such as cyclic hypoxia also need to be considered. Trials using localized modulation in the tumor microenvironment have the added advantage of minimizing off-target mutagenesis and systemic toxicity while preserving beneficial mutational processes in healthy cells.

#### 5.2.2. Increasing Response to Cancer Therapeutics

To date, many studies have focused on the anti-tumor benefits of APOBEC3 mutation activation, partly based on the observation that those patients with a higher mutation burden tended to respond better to immunotherapy, including the checkpoint inhibitors [76]. Overexpression of APOBEC3s was shown to increase responsiveness to targeted ATR (Ataxia Telangiectasia and Rad3-Related) and Chk1/2 (Checkpoint Kinase 1 and 2) inhibitors in acute myeloid leukemia and osteosarcoma cell lines [77,78]. Similarly, APOBEC3B overexpression sensitized tumor protein 53-deficient (p53-deficient) cells to Chk1/2 checkpoint kinases (regulate the cell cycle), Wee1 kinase (regulates the cell cycle), and PARP enzyme (involved in repairing breaks in ssDNA) inhibition [79]. Increased sensitivity to PARP inhibitors was also observed with APOBEC3A upregulation in pancreatic cancer cells [80]. In a 2018 study, Fujiki et al. [81] showed that APOBEC3B messenger RNA (mRNA) expression levels correlated with the efficacy of chemotherapy and that high APOBEC3B mRNA expression was a predictive factor for pathological Complete Response (pCR).

Research also indicates that members of the APOBEC3 family, particularly APOBEC3A and APOBEC3B, can upregulate PD-L1 (Programmed Death-Ligand 1) expression and potentially enhance responses to immunotherapy. It was shown that APOBEC3B upregulation is significantly associated with immune gene expression, including PD-L1 expression and T-cell infiltration in non-small cell lung cancer (NSCLC) [66]. Another study reported that APOBEC3A expression positively correlates with PD-L1 levels in many other cancers, including lung adenocarcinoma, urothelial carcinoma, breast invasive cancer, cervical cancer, and head and neck squamous cell carcinoma [41]. The study proposed that APOBEC3A induces PD-L1 expression through the c-Jun N-terminal kinase (JNK/c-JUN) signaling pathway, that is independent of interferon signaling.

However, there are also some studies reporting that inhibiting the APOBEC3 deaminases could be therapeutically beneficial. For example, it was reported that APOBEC3 inhibitors could prevent non-muscle invasive bladder cancer from progressing to muscle-invasive disease [82]. It was shown that APOBEC3B expression is associated with poor prognosis for breast cancer and some other cancers [83]. In a bioinformatics study by Ma et al. [84], it was shown that the high expression of APOBEC3G was also significantly associated with short overall survival (OS) in non-M3 acute myeloid leukemia (non-AML) patients, who are the vast majority of AML patients. This study also identified that treatment with crotonoside (a natural plant product) can reduce the expression of APOBEC3G and thus inhibit the viability of different AML cells in vitro. Crotonoside, a potent guanosine tyrosine kinase inhibitor with immunosuppressive effects, is now considered to be one possible natural candidate for APOBEC3G inhibition in non-AML patients.

Thus, while APOBEC3 inhibition holds therapeutic potential to improve patient outcomes in certain cancers by limiting tumor evolution, subclone heterogeneity, and therapy resistance, there may also be cases where APOBEC3 supplementation could enhance immune support and improve responsiveness to selected therapies. In both scenarios, the relationships between the different APOBEC3 isomers need to be better understood.

#### 5.2.3. Overcoming Drug Resistance

Acquired drug resistance to anticancer targeted therapies involving the APOBEC3 deaminases remains an unaddressed clinical challenge. However, there are studies reporting the association between the expression of particular members of the APOBEC3 family, and the development of drug resistance. In a recent study, it was shown that deletion of APOBEC3A reduces the number of mutations and the number of structural variations in persister cells and therefore delaying the development of drug resistance [85]. Thus, the suppression of APOBEC3A activity may represent a potential therapeutic strategy for preventing or delaying resistance to targeted therapies in lung cancer patients. Caswell et al. [86] showed that APOBEC3B could also be a useful target for overcoming NSCL immunotherapy drug resistance. APOBEC3B is known to promote tamoxifen resistance in estrogen receptor-positive (ER+) breast cancer patients [87]. Law et al. [87] also showed that APOBEC3B depletion in an ER+ breast cancer cell line results in prolonged tamoxifen responses in these murine xenograft experiments. These and several other studies suggest that multiple APOBEC3 family members contribute to targeted therapy resistance and therefore might be inhibited to overcome drug resistance [47,85].

#### 5.2.4. APOBEC3s and Cancer Progression

The first indication that changes in both adenosine and cytosine deaminase mutation signatures might be used to predict cancer progression was reported for a study of high-grade ovarian cancer (HGS-OvCa) in 2016 [88]. It has since been observed that APOBEC3B expression in various cancers, including breast, lung, and cervical cancers, leads to DNA mutations predicting progression [89]. Evidence showing that mutational changes in tumor phylogenies are associated with cancer progression has also been reported [57]. It is inferred from these studies that, at some stage during oncogenesis, some deaminases ‘self-edit’ and some activate alternate deaminase DBDs to generate the new mutation signatures observed. If these new DBD variants are linked to aggressive cancer, then they may become new therapeutic targets for suppressing or slowing cancer progression.

#### 5.2.5. APOBEC3 Modulation Drug Development Approaches

It is evident that both APOBEC3-enhancing and APOBEC3-inhibiting drug developments are already creating important new treatment options as early APOBEC3 deaminase modulation therapies develop.

Efforts to develop small molecule APOBEC3 inhibitors that target catalytic pockets of the APOBEC3s will help to guide the future design of inhibitors specific to each APOBEC enzyme [90,91]. Other potential strategies to reduce APOBEC3 activity include gene-silencing therapies and alternative splicing modulators [92]. For APOBEC3A, a nucleic acid secondary structure has emerged as a factor in determining substrate-targeting affinity, with a preference for ssDNA that forms stem–loop hairpin structures. Serrano et al. [93] used this knowledge to develop a nucleic acid-based inhibitor of APOBEC3A that showed specificity against APOBEC3A relative to the closely related catalytic domain of APOBEC3B. This work demonstrates the feasibility of leveraging secondary ssDNA structural preferences to design effective DNA inhibitors as potential therapeutics to inhibit APOBEC-driven viral and tumor evolution, and drug resistance. At the same time, we need to understand the clinical benefits of increasing APOBEC activity in cancers, particularly in the immunosuppressive hypoxic regions of tumors, and to increase the efficacy of immune checkpoint blockade therapy due to increased neoepitope presentation [94].

## 6. Adenosine Deaminase Modulation

Dysregulation of ADAR1 and ADAR2 deaminase A-to-I editing is now implicated in a wide range of diseases, including immune and inflammatory illness, neurological conditions (including schizophrenia), viral infections, and cancers [95,96]. It is also important to note that ADAR1,2 expression levels are not always correlated with the editing frequency indicating that there is another layer of factors modulating the editing frequency of ADARs and hence influencing the accumulated cell damage [97]. RNA editing can also modify some gene products without causing permanent changes in the genome and therefore has great potential in gene therapy [98]. Understanding how these complex layers of differential ADAR modulation influence the course of human pathologies is important for the development of future adenosine modulation therapeutics.

### 6.1. ADAR1 Tumor Promotion

Mutations caused by ADAR1 have been implicated in oncogenesis by contributing to tumor development and progression in various cancers [99,100]. In an ADAR1 study of thyroid cancer patients it was found that inhibiting ADAR1 profoundly repressed proliferation, invasion, and migration in thyroid tumor cell models [101]. In an early thyroid cancer study, it was shown that the pharmacological inhibition of ADAR1 activity with 8-azaadenosine reduced cancer cell aggressiveness [102]. However, it was subsequently found that 8-azaadenosine is not suitable for therapies that require selective inhibition of one ADAR isoform over another and may cause cellular toxicity likely due to broader disruptions in RNA metabolism [103]. An interesting study by Wang X et al. [104] has shown that the small-molecule ADAR1 inhibitor ZYS-1 can dramatically suppress prostate cancer cell growth and inhibit metastasis, and that it has a favorable safety profile. As most prostate cancer patients eventually relapse, these results identify ADAR1 as a potential druggable target using inhibition therapies such as ZYS-1. ADAR1 is also a potential druggable target in some breast cancers. Triple-negative breast cancers (TNBCs) are aggressive, chemotherapy-resistant, and have a poor prognosis. Baker et al. [105] showed that ADAR1 loss in TNBC cells resulted in reduced growth, reduced invasion, and diminished mammosphere formation, implying that ADAR1 helps promote these aggressive behaviors in TNBC. Consistent with this view, Binothman et al. [106] reported that ADAR1 is more abundantly expressed in invasive breast cancer (BC) tumors than in benign tumors.

Some studies have also identified ADAR1 as a potential druggable target to overcome resistance to therapy. In one study, it was shown that loss of ADAR1 in tumors overcomes resistance to immune checkpoint blockade [107]. This study identified the possibility that ADAR1 inhibitors could restore melanoma differentiation-associated protein 5 (MDA5), mitochondrial antiviral-signaling (MAVS) protein and IFN (MDA5-MAVS-IFN) signaling and inflammatory responses in tumors and resurrect their response to immune checkpoint blockade therapy. However, a study by Shiromoto et al. [108] suggested that the suppression of ADAR1 editing activity also resulted in genome instability and apoptosis, particularly in non-alternate lengthening of telomeres (non-ALT) and telomerase-positive cancers that are 70–80% of all types of cancers. Shiromoto predicted that ADAR1 inhibitors could be an effective treatment for cancer patients as they interfere with two completely different pro-oncogenic functions: suppression of MDA5-MAVS-IFN signaling by the cytoplasmic ADAR1p150, and maintenance of telomere stability in telomerase-reactivated cancer cells by the nuclear ADAR1p110 [108].

Dysregulation of the ADAR1 editing function has also been implicated in various autoimmune diseases. For instance, loss-of-function mutations in the ADAR1 gene are associated with Aicardi–Goutières syndrome (AGS), a congenital autoimmune disorder [109]. Loss of this ADAR1p150 function has been shown to cause embryonic lethality in ADAR1 null mice, and the severe autoimmune disease AGS in humans, and has been associated with resistance to immune checkpoint blockade in cancers [108]. Notably, ADAR1p150 binds not only right-handed (A-form) dsRNA, but also the left-handed Z-RNA duplex. Endogenous Z-RNAs arising from retroelements in mammalian genomes, if not edited and quenched by ADAR1p150, activate the innate immune sensor Z-NA Binding Protein 1 (ZBP1) and trigger ZBP1-dependent cell death. Such cell death drives autoimmunity in ADAR-deficient mice and may contribute to the development of AGS in humans. Other research indicates that adequate ADAR1 editing may serve as a defense mechanism against autoimmune diseases such as multiple sclerosis [110]. Yet significant overexpression of ADAR1 was found in rheumatoid arthritis (RA) synovial tissue and in the blood of patients with active RA [111], suggesting that ADAR1 could be a potential therapeutic target. Elevated ADAR1 expression and heightened A-to-I RNA editing activity is also a prominent feature of progressing HCC [112], gastric cancer [113] and a significant subset of progressing lung adenocarcinomas [114].

These examples underscore the importance of balancing A-to-I RNA editing in maintaining immune homeostasis and preventing unwanted oncogenic and autoimmune pathologies. The results also emphasize the potential applicability of ADAR1 inhibitors or ZBP1 agonists for several cancers.

### 6.2. ADAR2 Immunomodulation

Corresponding to the largely oncogenic role of ADAR1, there are now several studies reporting that ADAR2 plays a tumor-suppressive role in multiple cancers and in modulating the inflammatory responses.

In glioblastoma ADAR2 has been shown to slow progression by editing and modulating the function of the *GLI1* gene (Glioma-Associated Oncogene Homolog 1), which is a transcription factor and a key component of the Hedgehog (Hh) signaling pathway, which plays a crucial role in cancer progression [115]. Additionally, loss of ADAR2 in glioblastoma promotes tumor growth and resistance to therapy. In another glioblastoma study, restoration of ADAR2 editing activity has been shown to prevent tumor growth [116]. Further, it has been shown that the rescue of ADAR2 activity in glioblastoma cancer cells recovers the edited micro-RNA (miRNA) population lost in glioblastoma cell lines and tissues and rebalances expression of onco-miRNAs and tumor suppressor miRNAs to the levels observed in normal human brain [117]. Thus, ADAR2 largely reduces the expression of many miRNAs, most of which are onco-miRNAs. Similarly, in esophageal squamous cell carcinoma (ESCC), reduced levels of ADAR2 have been associated with poor prognosis as ADAR2 functions as a tumor suppressor in many cancers [112,113,118], including in hepatocellular and gastric cancers. Given the poor prognosis for these cancers, the restoration or supplementation of ADAR2’s function could provide therapeutic benefit.

ADAR2 is also found to lower the risk of autoimmune diseases by playing an important role in modulating inflammatory responses and thus lowering the risk of autoimmune diseases. A study on Borna disease virus (BoDV) demonstrated that ADAR2 edits the viral RNA, allowing it to mimic ‘self’ RNA [119]. This editing prevents the activation of innate immune responses. The study also showed that ADAR2-mediated RNA editing is essential for distinguishing self from non-self RNA and thereby reducing the risk of autoimmune responses. Deficient ADAR2 activity has been observed in amyotrophic lateral sclerosis (ALS) patients, resulting in improper *GluA2* gene editing [120]. These results are consistent with an earlier study reporting that mice lacking ADAR2 exhibit subsequent neurodegeneration [121]. Such studies underscore ADAR2’s essential role in neuronal survival, and there have been several studies exploring ADAR2 supplementation. For instance, a hyperactive ADAR2 variant capable of enhanced editing at specific RNA motifs has been identified [122]. Guide RNAs have also been designed to recruit endogenous ADAR2 to recode a loss-of-function mutation in the *PINK1* gene associated with Parkinson’s disease [123]. This approach successfully restored *PINK1* function in cellular models without introducing artificial proteins. Additionally, studies have shown that ADAR2 protein levels correlate with patient outcomes in glioblastoma multiforme (GBM) [124]. These studies highlight the therapeutic potential of ADAR2 supplementation and engineering in advancing the development of RNA-based therapies. However, as of now, there pre-clinical studies but no therapeutic interventions involving ADAR2 supplementation.

## 7. Harnessing the Power of Deaminase Modulation

There are a growing number of companies and research institutions that are part of a broader trend developing new biotechnologies to harness the power of deaminase modulation (Table 1). While there are several Adenosine deaminase (ADA) and Cytidine deaminase (CDA) modulating drugs on the market with proven therapeutic benefit, these are targeting metabolic enzymes. For example, the CDA cedazuridine is approved for clinical use in many countries. It has been shown to benefit patients with Myelodysplastic Syndromes (MDS), Chronic Myelomonocytic Leukemia (CMML), and Acute Myeloid Leukemia (AML) by enabling effective oral delivery of the hypomethylating agent decitabine. CDA normally degrades decitabine in the gut and liver, preventing adequate systemic exposure [125]. Cedazuridine inhibits CDA, allowing oral decitabine-cedazuridine to achieve systemic drug levels equivalent to intravenous decitabine. This pharmacokinetic equivalence translates to comparable clinical efficacy and safety while offering a more convenient, less burdensome oral treatment option for patients with these hematologic malignancies. Pentostatin (also known as deoxycoformycin) was developed by Pfizer Inc. (originally Parke-Davis) as an ADA catalytic inhibitor and is approved for treatment of hairy cell leukemia.

Table 1 also includes some of the companies at the forefront of developing base editing tools. These rely upon the precise editing by cytidine or adenosine deaminase base editors to introduce targeted mutations without creating double-strand breaks (DSBs) like clustered regularly interspaced short palindromic repeats (CRISPR)-Cas9 does. The Yu et al. (2020) [126] study describes how improved APOBEC-based cytosine base editors (CBEs) and adenine base editors (ABEs) can benefit disorders such as sickle cell disease (SCD), Fanconi anemia (FA), and cancers by enabling highly precise single-base corrections with minimal off-target activity. Some of these are now in pre-clinical or clinical trials. The authors engineered next-generation deaminase-based editors that substantially reduce unguided DNA and RNA deamination—a major limitation of earlier APOBEC-derived CBEs—while maintaining strong on-target editing efficiency [126]. This increased precision expands the therapeutic potential of base editing for several diseases driven by single-nucleotide mutations, or rewriting driver or resistance-associated mutations in cancer. Tumors often exploit ADAR1, an RNA-editing enzyme, to alter double-stranded RNA and suppress innate immune sensing, thereby avoiding T cell-mediated clearance. Rebecsinib has been developed by Accent Therapeutics (with AstraZeneca) as an agonist of the ADAR1p150 isoform. It has been shown to counteract ADAR1-mediated immune evasion in solid tumors [127]. Rebecsinib restores normal ADAR1 splicing and enhances ADAR1p150 activity, reactivating immune recognition of tumor cells. By reversing the malignant splice isoform switch, this approach promotes anti-tumor immunity, potentially improving patient responses to immunotherapies in cancers where ADAR1-driven immune evasion contributes to therapy resistance. Development of Rebecsinib is now in pre-clinical trials and approved as an Investigational New Drug (IND) for use in human trials. In another example, using ADAR2-mediated RNA editing, Korro Bio is developing KRRO-110 in its Phase 1/2a REWRITE trial to treat patients with alpha-1 antitrypsin deficiency (AATD). The therapy leverages ADAR2 to precisely edit RNA transcripts, correcting pathogenic mutations in the SERPINA1 gene responsible for producing dysfunctional alpha-1 antitrypsin protein. By restoring the expression of functional protein at the RNA level, this approach aims to reduce the accumulation of misfolded protein in the liver and improve systemic levels of functional alpha-1 antitrypsin, addressing both liver and lung disease manifestations of AATD in a targeted, potentially safer manner than traditional gene therapy. Wave Life Sciences have also advanced RNA editing to Clinical Phase 1/2a trial. They have developed the WVE-006 that uses targeted A→I RNA editing to repair the single-nucleotide Z mutation in *SERPINA1* mRNA, enabling AATD (Pi*ZZ) patients to produce functional wild-type M-AAT protein rather than the misfolded, liver-toxic Z-AAT variant. In their Phase 1/2a RestorAATion trials (NCT06186492, NCT06405633), subcutaneous, GalNAc-delivered WVE-006 has shown dose-dependent in-human RNA editing, restoring circulating M-AAT to therapeutically relevant levels (~10–12 µM) and increasing neutrophil-elastase inhibitory activity, indicating that the newly produced protein is functional [128]. By both boosting protective AAT for the lungs and potentially reducing hepatocyte stress from Z-AAT accumulation, WVE-006 offers a dual-organ disease-modifying strategy that could meaningfully improve outcomes for AATD patients if durability and clinical endpoints are confirmed in ongoing cohorts.

Thus, several promising base editing therapies are now advancing through clinical development. It is also likely that many other developments remain unpublished to prevent competitors from copying, or to secure patents. There are also several practical challenges that need to be addressed to support wider clinical translation. Firstly, regulatory approval requires careful attention to safety as deaminase activity can produce unintended edits in both DNA and RNA and can engage innate immune pathways [129]. Off-target DNA and RNA edits have been observed with engineered deaminases and base editors. These can range from single-base substitutions at unintended genomic loci to transcriptome-wide RNA editing events and may increase mutational burden or perturb gene expression [129]. There may also be some unknown in vivo off-target outcomes. For example, because endogenous deaminases have physiologic roles in immunity and RNA/DNA metabolism, perturbing their expression or activity could have complex downstream effects such as altering immune signaling or causing genome instability [130]. An additional concern is determining the dose titration balance between therapeutic potency and specificity [131]. Efficient, tissue-specific delivery of deaminase-modulating editing machinery or oligonucleotides also remains a major bottleneck for many indications. Viral vectors, lipid nanoparticles, and conjugated oligonucleotides each have trade-offs in tropism, payload size, durability, and immune activation [132]. Thus, the early-phase trials or healthy-volunteer studies being conducted will be important for defining the balance between these efficacy and safety concerns.

Given the complexities associated with modulation of the key endogenous human deaminases, Figure 3 provides a summary of when activation or inhibition of deaminases is either beneficial or harmful for different diseases.

**Table 1 ijms-26-11532-t001:** List of organizations at the forefront of developing deaminase modulation drugs and drug development tools.

Organization	Deaminase(s)	Modulation Type	Disease Indication(s)	Phase	Notes; Source/[References]
Accent Therapeutics with AstraZeneca, Lexington, MA, USA	ADAR1	Inhibition	Lung, breast, ovarian, head & neck cancers	Pre-clinical	Developing small-molecule ADAR1 inhibitors. Citations: [127]; company pipeline.
AIRNA Inc., Cambridge, MA, USA	ADAR2	RNA editing	Genetic diseases	Pre-clinical	ADAR-recruiting oligonucleotides. Source: AIRNA pipeline.
ApoGen Biotechnologies, Seattle, WA, USA	APOBEC3 family	Inhibition	APOBEC-driven mutagenic cancers	Pre-clinical	APOBEC3 inhibitors for reducing tumor mutational burden. Source: ApoGen investor materials.
Arsenal Biosciences, South San Francisco, CA, USA	Cytidine/adenosine deaminases	Base editing for T-cell engineering	Solid tumors	Pre-clinical	T-cell engineering using base editors. https://arsenalbio.com/2022/01/27/computational-biologys-effect-on-solid-tumors/ (accessed on 19 November 2025)
Aspera Biomedicines, La Jolla, CA, USA	ADAR1 (p150)	Inhibition	Solid tumors with ADAR1-mediated immune evasion	Pre-clinical/IND-enabling	Rebecsinib ADAR1p150 antagonist. Citation: [127].
Astex Pharmaceuticals, Pleasanton, CA, USA	Cytidine deaminase	Inhibition	MDS, CMML, AML	Approved	Cedazuridine. https://www.fda.gov/drugs/resources-information-approved-drugs/fda-approves-oral-combination-decitabine-and-cedazuridine-myelodysplastic-syndromes? (accessed on 19 November 2025)
Beam Therapeutics, Cambridge, MA, USA	APOBEC-based CBEs, ABEs	Base editing	Sickle cell, FA, oncology	Clinical + Pre-clinical	CBE/ABE programs. Citation: [126].
Bio-Techne (Tocris), Minneapolis, MN, USA	Adenosine deaminase (ADA)	Inhibition	Immunomodulation/research use	Pre-clinical	EHNA hydrochloride ADA inhibitor. Source: Tocris datasheet.
Broad Institute/MIT/Harvard, MA, USA	Engineered cytidine deaminase	RNA C→U editing	RNA diseases	Pre-clinical	RESCUE C→U RNA editing system.
Covant Therapeutics with Boehringer Ingelheim, Boston, MA, USA	ADAR1	Inhibition	Immunotherapy-enhanced cancers	Pre-clinical	Covalent ADAR1 inhibitors. Source: company pipeline releases.
Editas Medicine, Cambridge, MA, USA	Multiple deaminases	Base editing	Genetic diseases	Pre-clinical	Deaminase-Cas fusion editors.
GMDx Genomics, Melbourne Australia	Multiple deaminases	Bioinformatic mapping	Precision oncology	Research	Deaminase binding domain definition. Citation: [1].
Halozyme Therapeutics, San Diego, CA, USA	ADA2	Enzyme supplementation	Colon, lung, pancreatic cancers	Pre-clinical	PEG-ADA2 reduces tumor adenosine. Source: Halozyme preclinical publications.
HuidaGene Therapeutics, Shanghai, China	Novel guanine deaminase/synthetic editors	G→C or G→T editing	Genetic disorders	Pre-clinical	World’s first G→Y base editor. Citation: [133].
Korro Bio, Cambridge, MA, USA	ADAR2	RNA editing	AATD	Clinical (Phase 1/2a)	KRRO-110 (REWRITE trial).
Life Edit Therapeutics, Morrisville, NC, USA	Multiple deaminases	Base editing	Genetic diseases	Pre-clinical	In vivo mRNA base editing. Company platform sources.
Mammoth Biosciences, Brisbane, CA, USA	Engineered deaminases	Base editing	Genetic & infectious disease	Pre-clinical	Compact Cas-based deaminase editors.
Pfizer Inc., Detroit, MI, USA	Adenosine deaminase (ADA)	Inhibition	Hairy cell leukemia	Approved	Pentostatin (also known as deoxycoformycin). Source Pfizer product information.
Scribe Therapeutics, Alameda, CA, USA	Engineered deaminases	Base editing	Neurological & genetic disease	Pre-clinical	CRISPR-based precision deaminases. Citation: [126].
Shape Therapeutics, San Francisco Bay, CA, USA	ADAR2	RNA editing	Genetic disease	Pre-clinical	ADAR2-based RNA editing via AAV. Source: ShapeTx platform.
UCSF RBVI, San Francisco Bay, CA, USA	Multiple deaminases	Molecular modeling	Research	Research tools	Chimera/ChimeraX platform.
University of Reading, UK	Multiple deaminases	Protein modeling	Research	Research tools	IntFOLD protein prediction tools.
Wave Life Sciences, Cambridge, MA, USA	ADAR	A→I RNA editing	AATD	Clinical (Phase 1/2a)	WVE-006 clinical trials NCT06186492, NCT06405633. Source: News Release 16 October 2024.
WHAT IF Foundation, Nijmegen, The Netherlands	Multiple deaminases	Structural modeling	Research	Research tools	3D enzyme–motif modeling. Source: WHAT IF documentation.

Acronyms: AATD—Alpha-1 Antitrypsin Deficiency; AAV—Adeno-Associated Virus; ABE (in APOBEC-based ABEs)—Adenine Base Editor; ADA—Adenosine Deaminase; AML—Acute Myeloid Leukemia; APOBEC-based CBEs—APOBEC-based Cytidine Base Editors; CMML—Chronic Myelomonocytic Leukemia; CRISPR—Clustered Regularly Interspaced Short Palindromic Repeats; EHNA hydrochloride—Erythro-9-(2-Hydroxy-3-Nonyl) Adenine Hydrochloride (an ADA inhibitor); IND—Investigational New Drug (FDA) term for new drug able to be tested on humans); KRRO-110—KRRO-110 (small-molecule ADAR1 inhibitor; spelling sometimes KRRO110); *LAG-3*—Lymphocyte Activation Gene-3; MDS—Myelodysplastic Syndromes; NCT06186492—Clinical trial identifier (NCT number) for a study of WVE-006 for AATD; PEG-ADA2—Pegylated Adenosine Deaminase Type 2; RESCUE—RNA Editing for Specific C to U Exchange (an ADAR-based therapeutic RNA-editing platform); WVE-006—WVE-006 (Wave Life Sciences ADAR-mediated RNA-editing therapeutic for AATD).

## 8. Closing Remarks

A new area of drug development based on deaminase modulation is gaining traction because it has broad therapeutic application in oncology, immunology, virology, infectious diseases, neurology and gene editing. With ongoing clinical trials and rapid biotechnological advancements, this new era in drug development is likely to meaningfully improve clinical outcomes. However, while several companies are already developing technologies and drugs that leverage the potential of deaminase modulation, we are still several years away from being able to deliver the wide range of promised therapeutic benefits. Because deaminases are natural and highly targeted mutagenic enzymes that play crucial roles in immunity and the progression of all inflammation-linked diseases, they represent promising targets for advancing the next generation of therapeutic development.

## Figures and Tables

**Figure 1 ijms-26-11532-f001:**
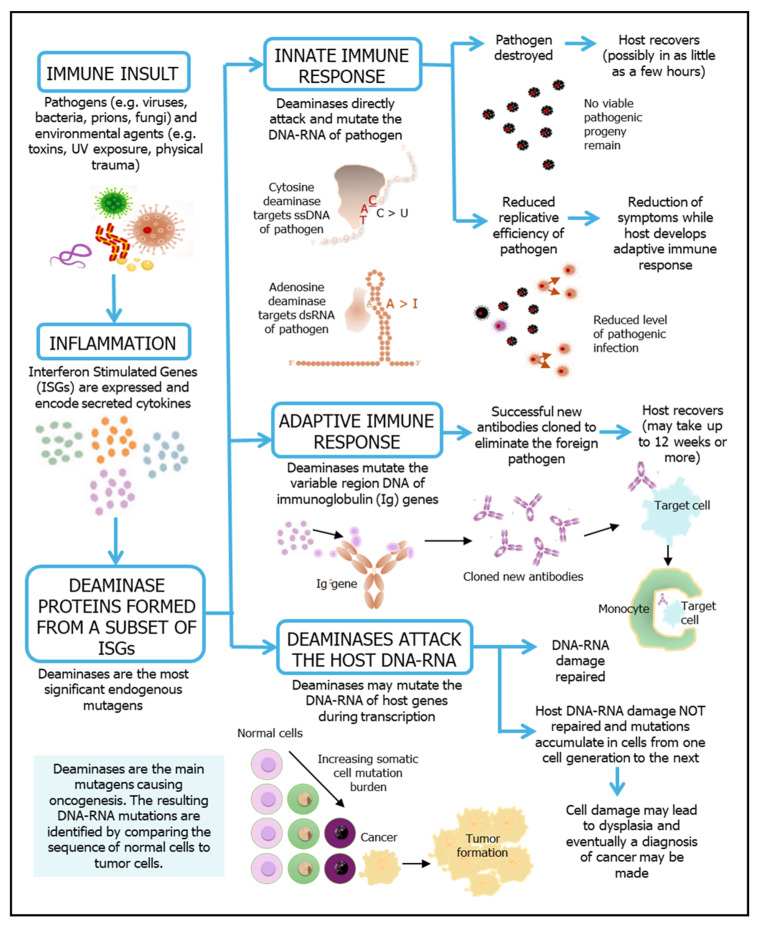
Diagram linking an immune insult with the roles of deaminases in immunity and disease. An immune insult triggers an inflammation response in the host. As a part of the inflammation response, deaminase proteins are expressed as a subset of the interferon stimulated genes (ISGs) [1,4]. Deaminase genes are transcribed and translated into proteins in affected cells. During an innate immune response, the deaminases primarily attack the ssDNA or dsRNA of the pathogen to either destroy the pathogen, or to reduce the number of viable progeny. During an adaptive immune response, the AID is essential for somatic hypermutation (SHM) and class switch recombination (CSR) for generating both the diversification of the functional class of immunoglobulins (Ig) and by enhancing antibody specificity and changing affinity. The release of deaminase proteins may also result in uncorrected somatic mutations during cellular transcription. The resulting DNA-RNA damage may be accumulated from one cell generation to the next and could ultimately give rise to inflammation-linked disease progression, or cancer.

**Figure 2 ijms-26-11532-f002:**
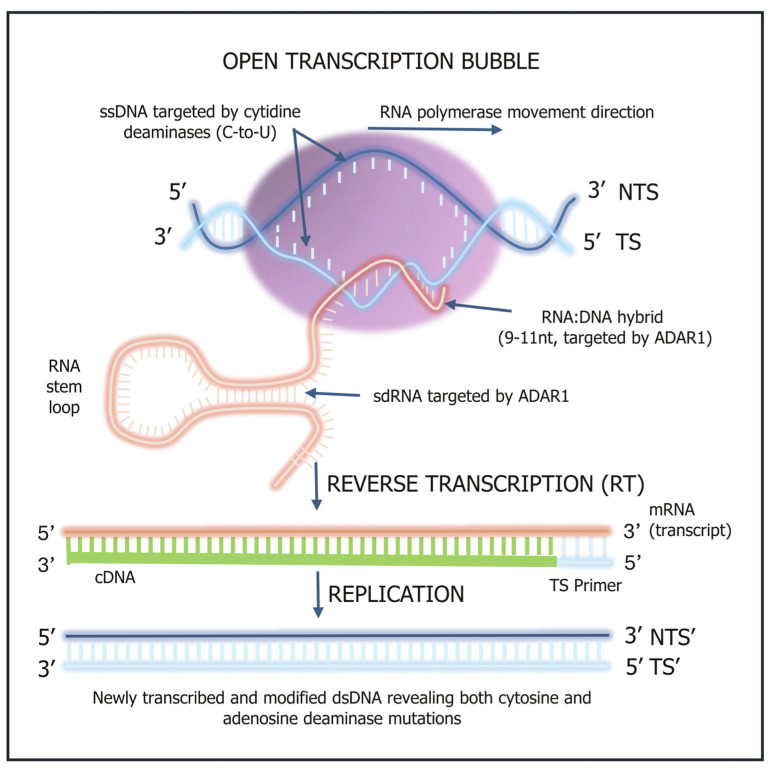
Schematic of the key cellular steps involved in the deaminase-driven reverse transcription (DRT) model. When single stranded DNA (ssDNA), double stranded RNA (dsRNA) and the 9–11 nucleotide RNA:DNA hybrid structures are formed, these become accessible targets for deaminases [5,6]. The NTS is the non-transcribed strand and TS refers to the transcribed strand. The open transcription bubble provides ssDNA that may be deaminated by the cytidine deaminases AID and APOBECs mediating C-to-U modifications. APOBEC3A may also deaminate nascent ssRNA. The RNA:DNA hybrid and the RNA stem loop conformations provide dsRNA that is targeted by transcription-linked adenosine deaminases acting on RNA (ADAR1). ADAR1-mediated A-to-I modifications of adenines in the nascent RNA and DNA at RNA:DNA hybrids occur in progressing cancer cells under replicative and transcriptional stress [8,9]. A cellular reverse transcription (RT) step results in the formation of complementary DNA (cDNA) from the copying of the newly synthesized mRNA template as shown, which is then replicated producing a new NTS (NTS’) and TS (TS’). The TS’ is used to identify the signatures of any new uncorrected mutations caused by both the cytidine (C) deaminases (AID and APOBECs) as well as the adenosine (A) deaminases (ADARs 1 and possibly 2). For a more detailed molecular description of the DRT model and other mutagenic catalyzes that may result in some additional mutation signatures refer to Steele and Lindley [6].

**Figure 3 ijms-26-11532-f003:**
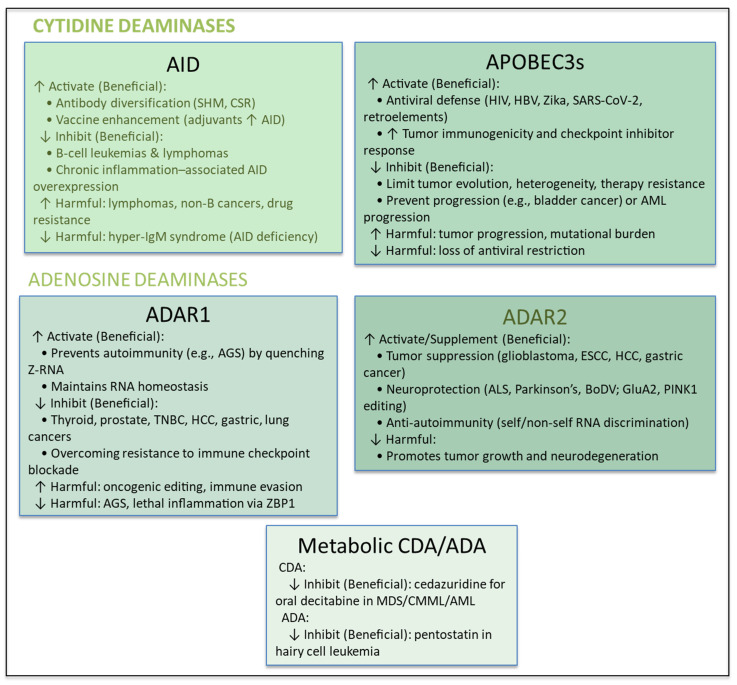
A summary of when activation (up arrow) or inhibition (down arrow) of deaminases is either beneficial or harmful for different diseases. Key theme for the above is that activation is beneficial when boosting immunity or tumor suppression; inhibition is beneficial when deaminase-driven mutagenesis fuels cancer or drug resistance. Balance is critical to avoid autoimmunity, genome instability, or loss of host defense. Acronyms: ADA—Adenosine Deaminase; AGS—Aicardi–Goutières Syndrome; AML—Acute Myeloid Leukemia; BoDV—Borna Disease Virus; CDA—Cytidine Deaminase; CMML—Chronic Myelomonocytic Leukemia; CSR—Class Switch Recombination; ESCH—(Assumed: Esophageal Squamous Cell Carcinoma; commonly “ESCC”); GluA2—Glutamate Ionotropic Receptor AMPA Type Subunit 2; HBV—Hepatitis B Virus; HCC—Hepatocellular Carcinoma; HIV—Human Immunodeficiency Virus; IgM—Immunoglobulin M; MDS—Myelodysplastic Syndromes; PINK1—PTEN-Induced Putative Kinase 1; SARS-CoV-2—Severe Acute Respiratory Syndrome Coronavirus 2; SHM—Somatic Hypermutation; TNBC—Triple-Negative Breast Cancer; ZBP1—Z-DNA Binding Protein 1; Z-RNA—Left-handed Z-conformation RNA.

## Data Availability

No new data were created or analyzed in this study. Data sharing is not applicable to this article.

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
