# Peer review of "Deaminase Modulation Driving a New Era in Drug Development"

_ijms, 2025, doi:10.3390/ijms262311532_

Round 1
Reviewer 1 Report
Comments and Suggestions for Authors
The manuscript entitled “Deaminase Modulation Driving a New Era in Drug Development” is within the scope of IJMS and could be accepted but after minor revision.
The review needs to be enriched with more examples of deaminase modulating drugs either in Clinical use or under Clinical trials
.
Author Response
Reviewer 1: Enrich with more examples of deaminase modulating drugs in Clinical Use or under Clinical Trial.
Please refer to an Updated Table 1. All deaminase modulation drugs included in Table 1 that are approved for clinical use, are in Clinical Trials (Phase 1/2a), or are approved as an Investigational New Drug (IND) for use in human trials (FDA term) are briefly described in the main text. A brief description of the deaminase modulation type and activity, the main disease indication(s) and reference source(s) are included. It is noted that most of the deaminase drugs in Clinical Use or under Clinical Trial are categorized as: Adenosine deaminase (ADA) and Cytidine deaminase (CDA) modulating drugs; or, cytosine base editors (CBEs) and adenine base editors (ABEs). These are described in an expanded Section 7 ‘Harnessing the power of deaminase modulation’.
There are also several practical challenges that need to be addressed to support wider clinical translation of the other deaminase modulation drugs in developed and described in the main text. These are also briefly described in Section 7.
Reviewer 2 Report
Comments and Suggestions for Authors
The review by Lindley provides an enthusiastic and wide-ranging overview of deaminase modulation and its potential in drug development.
The author’s deep familiarity with the field is evident.
However, the following points should be addressed:
- Please add a figure summarizing when activation or inhibition of deaminases is beneficial or harmful in different diseases.
- Please move Table S1 into the main text and expand it to include the specific deaminase target, modulation type (activation or inhibition), disease indication, and development phase.
- Please temper general hyperbolic statements such as “revolutionize medicine.”
- Please add a brief discussion on safety, off-target mutagenesis, and current limitations of deaminase-based therapies.
- Please clarify the authorship, as the extensive acknowledgments suggest that some contributors may warrant co-authorship.
Recommendation: Accept with major revision.
Author Response
Reviewer 2: However, the following points should be addressed:
- Please add a figure summarizing when activation or inhibition of deaminases is beneficial or harmful in different diseases.
See Figure 3 added as requested. I think that this figure greatly enhances the main message of the paper. Thank you.
2. Please move Table S1 into the main text and expand it to include the specific deaminase target, modulation type (activation or inhibition), disease indication, and development phase.
Table S1 now integrated into the main text as Table 1, and it includes the added field breakdown requested. Acronyms and full Reference citations are included.
3. Please temper general hyperbolic statements such as “revolutionize medicine.”
In the Concluding Remarks section, I have made the following changes:
- “revolutionize medicine” – replaced with “meaningfully improve clinical outcomes”
- “The fact that the deaminases are natural and highly targeted mutagenic enzymes that play crucial roles in immunity and the progression of all inflammation-linked diseases makes them powerful agents for forging this new frontier in drug development.” – sentence rewritten as “Because deaminases are endogenous, highly targeted mutagenic enzymes with essential roles in immunity and in the progression of inflammation-associated diseases, they represent promising targets for advancing the next generation of therapeutic development”
I did not identify any other hyperbolic terminology in the main text. However, I have also rewritten the Abstract to eliminate hyperbolic statements and impressions as follows:
Submitted Abstract: Our growing knowledge of the complex roles of the endogenous mutagenic deaminases in human disease is fueling the development of a fundamentally new generation of drugs that are likely to revolutionize medicine. These new drugs and drug development opportunities are designed to harness therapeutic benefits by modulating deaminase behavior. The fact that the deaminases are endogenous enzymes playing crucial roles in inflammation-linked diseases makes them powerful agents for forging this new frontier in drug development. While only a few deaminase modulating drugs are approved for clinical use, many are in development. We provide examples to highlight how we can unlock the healing power harnessed by this amazing orchestra of enzymes. We also identify the challenges and new opportunities not currently being acted upon.
Revised Abstract: Our expanding understanding of the complex roles of endogenous mutagenic deaminases in human disease is driving the development of a new generation of therapeutics. These emerging drugs aim to achieve clinical benefit by modulating deaminase activity. Because these enzymes are intrinsic to key inflammation-related pathways, they represent promising targets for future therapeutic innovation. Although only a small number of deaminase-modulating agents have been approved for clinical use, many more are currently under investigation. Here, we present examples that illustrate the therapeutic potential of modulating this diverse family of enzymes and identify some of the challenges and opportunities that warrant further exploration.
4. Please add a brief discussion on safety, off-target mutagenesis, and current limitations of deaminase-based therapies.
This is a good point. I had included the points below in a previous version before I reduced the overall word count. I have edited previous text and inserted additional text in Section 7 - before Closing Remarks:
There are also several practical challenges that need to be addressed to support wider clinical translation. Firstly, regulatory approval requires careful attention to safety as deaminase activity can produce unintended edits in both DNA and RNA and can engage innate immune pathways [129]. Off-target DNA and RNA edits have been observed with engineered deaminases and base editors. These can range from single-base substitutions at unintended genomic loci to transcriptome-wide RNA editing events and may increase mutational burden or perturb gene expression [129]. There may also be some unknown in vivo off-target outcomes. For example, because endogenous deaminases have physiologic roles in immunity and RNA/DNA metabolism, perturbing their expression or activity could have complex downstream effects such as altering immune signaling or causing genome instability [130]. An additional concern is determining the dose titration balance between therapeutic potency and specificity [131]. Efficient, tissue-specific delivery of deaminase modulating editing machinery or oligonucleotides also remains a major bottleneck for many indications. Viral vectors, lipid nanoparticles, and conjugated oligonucleotides each have trade-offs in tropism, payload size, durability, and immune activation [132]. Thus, the early-phase trials or healthy-volunteer studies being conducted will be important for defining the balance between these efficacy and safety concerns.
5. Please clarify the authorship, as the extensive acknowledgments suggest that some contributors may warrant co-authorship.
Because of the broad range of fields that this article draws together, the persons named in the acknowledgements were requested to provide editorial advice in their particular field of expertise. This was done after the MS was drafted to ensure the accuracy of the content. I therefore do not believe, and they would all agree, that any of the editorial suggestions contributed by them warrant co-authorship.
Round 2
Reviewer 2 Report
Comments and Suggestions for Authors
The author has addressed my requests.